# Comparison of Deep Learning Algorithms in Predicting Expert Assessments of Pain Scores during Surgical Operations Using Analgesia Nociception Index

**DOI:** 10.3390/s22155496

**Published:** 2022-07-23

**Authors:** Wei-Horng Jean, Peter Sutikno, Shou-Zen Fan, Maysam F. Abbod, Jiann-Shing Shieh

**Affiliations:** 1Department of Mechanical Engineering, Yuan Ze University, Taoyuan 320, Taiwan; s1078702@mail.yzu.edu.tw (W.-H.J.); s1085022@mail.yzu.edu.tw (P.S.); 2Department of Anesthesiology, Far Eastern Memorial Hospital, Banqiao District, New Taipei City 220, Taiwan; 3Department of Anesthesiology, College of Medicine, National Taiwan University, Taipei 100, Taiwan; shouzen@ntuh.gov.tw; 4Department of Anesthesiology, En Chu Kong Hospital, New Taipei City 237, Taiwan; 5Department of Electronics and Electrical Engineering, Brunel University London, London UB8 3PH, UK

**Keywords:** surgical operation, analgesia nociception index (ANI), expert assessment of pain score (EAPS), multilayer perceptron (MLP), long short-term memory (LSTM)

## Abstract

There are many surgical operations performed daily in operation rooms worldwide. Adequate anesthesia is needed during an operation. Besides hypnosis, adequate analgesia is critical to prevent autonomic reactions. Clinical experience and vital signs are usually used to adjust the dosage of analgesics. Analgesia nociception index (ANI), which ranges from 0 to 100, is derived from heart rate variability (HRV) via electrocardiogram (ECG) signals, for pain evaluation in a non-invasive manner. It represents parasympathetic activity. In this study, we compared the performance of multilayer perceptron (MLP) and long short-term memory (LSTM) algorithms in predicting expert assessment of pain score (EAPS) based on patient′s HRV during surgery. The objective of this study was to analyze how deep learning models differed from the medical doctors′ predictions of EAPS. As the input and output features of the deep learning models, the opposites of ANI and EAPS were used. This study included 80 patients who underwent operations at National Taiwan University Hospital. Using MLP and LSTM, a holdout method was first applied to 60 training patients, 10 validation patients, and 10 testing patients. As compared to the LSTM model, which had a testing mean absolute error (MAE) of 2.633 ± 0.542, the MLP model had a testing MAE of 2.490 ± 0.522, with a more appropriate shape of its prediction curves. The model based on MLP was selected as the best. Using MLP, a seven-fold cross validation method was then applied. The first fold had the lowest testing MAE of 2.460 ± 0.634, while the overall MAE for the seven-fold cross validation method was 2.848 ± 0.308. In conclusion, HRV analysis using MLP algorithm had a good correlation with EAPS; therefore, it can play role as a continuous monitor to predict intraoperative pain levels, to assist physicians in adjusting analgesic agent dosage. Further studies may consider obtaining more input features, such as photoplethysmography (PPG) and other kinds of continuous variable, to improve the prediction performance.

## 1. Introduction

Since the introduction of ether as an anesthetic agent, surgical operations have shown great progress. Every day there are many surgical operations performed to treat patient diseases, for damaged tissue repair, or of excision tumors worldwide. Since almost all surgical procedures are painful, general anesthesia with adequate hypnosis, analgesia, and muscle relaxation is needed [1]. It is crucial for physicians to assess pain to decide certain treatment plans and to consider the effects of those plans. The assessment of pain can be either subjective or objective [2]. Under general anesthesia, surgical patients are unconscious, so pain will be assessed objectively.

Painful stimulation alters the autonomic nervous system, resulting in sympathetic activation and parasympathetic deactivation. Sympathetic activation increases heart rate and blood pressure. In the past, clinical experience and vital signs such as heart rate and blood pressure, were used to adjust analgesic dosages [3]. Continuously evaluating the intraoperative pain level of surgical patients in the operation room is crucial for a physician to determine whether the patients are in adequate anesthesia and analgesia.

Heart rate variability (HRV) is the variation over time of the period between consecutive heartbeats. HRV is expressed as the variation of RR interval derived from an electrocardiogram (ECG). As well as being a good indicator of overall cardiac health and autonomic nervous system (ANS) status, HRV can reveal how well the heart responds to certain unpredictable stimuli. ANS is in good health when the sympathetic nervous system (SNS) and parasympathetic nervous system (PNS) are in balance. In response to a certain stimulus, such as pain, SNS activity increases and the PNS activity decreases, causing an increase of the heart rate. HRV will decrease as the heart rate increases. A relaxed condition, however, results in a decreased SNS activity and an increased PNS activity. The HRV will increase as a result. Nowadays, there is the analgesia nociception index (ANI), based on electrocardiogram (ECG) signal measurements of heart rate variability (HRV). Commercial products for online pain monitoring have already incorporated this technology for evaluating pain non-invasively [4]. In clinical settings, ANI has been applied in laparoscopic abdominal surgery [1], spinal stabilization surgery [5], and acute postoperative pain evaluation [6]. Aside from surgical operations, it has also been used to assess parasympathetic changes related to emotional states [7].

Artificial intelligence (AI) is defined as “a field of computer science and engineering concerned with the computational understanding of what is commonly called intelligent behavior, and with the creation of artifacts that exhibit such behavior” [8]. Deep learning is a type of AI that has been widely applied in clinical fields. Applying it in the case of surgical operations, to assess surgical pain, is also possible. Surgical pain is usually evaluated according to medical doctors′ clinical knowledge and experience, and it usually takes a longer time. Although there are several commercialized tools, such as ANI monitoring devices, all clinical needs regarding pain prediction have not yet been fulfilled. The presence of deep learning algorithms with the ability to acquire, analyze, and apply a great deal of knowledge [9] will assist medical doctors in evaluating pain, so that any clinical decisions can be made earlier, based on the predicted expert assessment of pain score (EAPS) during surgical operation.

Multilayer perceptron (MLP) was developed based on biological nervous systems, especially the human brain, and has the ability to receive information, transmit it, process it, and finally output responses based on the information received. MLP consists of input, hidden, and output layers. The ability of MLP to deal with nonlinearity is one of its most important characteristics [10].

Long short-term memory (LSTM) is an algorithm that addresses the problem of the vanishing gradients in traditional recurrent neural network (RNN), thus enabling it to handle long-term dependencies. Recurrent connections in LSTM yield outputs that are mapped from both the current input and the input from the previous time steps [11,12]. LSTM consists of three gates: a forget gate, input gate, and output gate, which regulate the flow of information in the cell [13].

The evaluation of pain during surgical operations has been demonstrated in several previous studies. M. Jeanne et al. [1] evaluated the effect of ANI on heart rate (HR) and systolic blood pressure (SBP) under various noxious stimuli during laparoscopic abdominal surgery. Researchers found that ANI decreased significantly in the painful stimulus group, but HR and SBP did not change significantly, indicating that ANI was more sensitive for evaluating pain in propofol-anesthetized patients. M. Gruenewald et al. [3] examined several variables, including ANI, surgical pleth index (SPI), bispectral index (BIS), HR, and non-invasive mean arterial blood pressure (MAP) in patients undergoing elective surgery. ANI and SPI were significantly changed at all steps of tetanic stimulation. In propofol-remifentanil anesthesia, these two variables detected stimulation better than BIS, HR, and MAP. In a study conducted by Hans et al. [14], intravascular volume status was modified through a fluid challenge (FC) during neurosurgery. In this study, SPI responded differently to FC, depending on the intravascular volume status and propofol concentration. When FC occurred, SPI changes correlated with pulse wave amplitude, not heart rate.

There have been previous applications of deep learning algorithms in the medical field, including MLP and LSTM. Using fifteen types of vital signs, Hu et al. [15] developed an MLP model to predict clinical deterioration in hospitalized adult patients with hematologic malignancies. With an 82% positive predictive value, that MLP model outperformed VitalPac Early Warning Score (ViEWS) with a 24% positive predictive value. An MLP model was developed by Sadrawi et al. [16] using the sample entropy of continuous electroencephalogram (EEG) signals, electromyography (EMG), heart rate, pulse, systolic and diastolic blood pressure, and signal quality index (SQI) as the input features for predicting consciousness level which was scored by five medical doctors. With 10-fold cross validation, the MLP model showed a mean absolute error (MAE) of 6.61 ± 0.15 and outperformed the BIS value, which had an MAE of 12.31 ± 13.06. Sharifi and Alizadeh [17] developed an MLP model using twelve factors to predict chronic kidney disease. The proposed MLP model achieved a 98% accuracy. For predicting cardiac arrest and respiratory failure, Kim et al. [18] used vital signs, treatment history, recent surgical history, and current health status as the input features. For cardiac arrest and respiratory failure, the areas under the receiver operating characteristic curve of the LSTM model were 0.886 and 0.869, respectively. Through LSTM, Sideris et al. [19] predicted continuous arterial blood pressure (ABP) signals from the photoplethysmograms (PPGs) of intensive care unit (ICU) patients. The LSTM model resulted in an accurate estimation of SBP, with a root mean square error of 2.58 ± 1.23.

ANI, which is derived from HRV as the representation of parasympathetic activity, is a commercialized tool for evaluating pain. Although painful stimulation, in a theoretical context, will activate sympathetic activity and reduce parasympathetic activity, in reality, not only painful stimulation results in the activation of sympathetic activity. Other factors are also able to influence the sympathetic–parasympathetic balance, so in real applications at the hospital, evaluating pain only based on patient’s HRV is not enough. In this case, medical doctors also consider other factors besides HRV to determine EAPS, which is the true gold standard of pain scoring, and to adjust anesthetic dosage in the operation room. Therefore, considering their excellent performance in pattern recognition, which is required by medical doctors in the operation room to obtain a better prediction model, in our current research, we applied deep learning algorithms, such as MLP and LSTM, to predict the gold standard of pain score, instead of directly using ANI for evaluating pain level. These algorithms are the most suitable for application on the input features, which are in the form of time series data; whereas another type of deep learning algorithm, a convolutional neural network (CNN), is more suitable for application on image data. Moreover, a deep learning algorithm, with its activation function, is capable of dealing with non-linearity, so it is suitable for the regression task in the prediction of EAPS. Compared to deep learning algorithms, traditional machine learning algorithms are more suitable for use in classification and linear regression tasks. Since it does not support nonlinear regression, the prediction results only appeared in the form of a flat horizontal line and could not fluctuate at all when traditional machine learning algorithms were used in our research.

## 2. Materials and Method

### 2.1. Patients and ECG Signal

In 2015, 142 patients underwent surgical operations at National Taiwan University Hospital (NTUH). ECG signals were recorded using a Philips IntelliVue MP60 physiological signal monitoring device, at a 512-Hz sampling rate. A computer is used to store the recorded data from that device. In total, 62 patients were excluded: 35 patients′ surgical time was less than 60 min, 15 patients had improper shape of ECG signal, and 12 patients had incomplete or missing ECG signal and EAPS label data. Finally, 80 patients were included in this research. The surgical duration varied from one to five hours.

### 2.2. Expert Assessment of Pain Score (EAPS)

The nurses observed the entire surgical process and took notes regarding clinical events and signs, such as HR and ABP measurements, types of hypnotics and analgesics, patient′s movements, and unusual responses. The expert assessment of pain score (EAPS) was calculated separately after the surgical operation was finished by five medical doctors (Doctors A, B, C, D, and E) with professional experience in surgical operations. A higher EAPS indicates a more painful surgery. Based on their clinical knowledge and experience, these five doctors interpreted the notes or recordings provided by the nurses to determine the EAPS. The patients were not in contact with them in this case. EAPS curves were plotted on recording papers by hand. These papers were then scanned and digitized into numerical data, with one data point of EAPS for every second [20].

### 2.3. Calculation of ANI

The flowchart shown in Figure 1 summarizes the steps for calculating ANI. As the first step, the R peaks of the ECG signal were detected, in order to obtain the RR interval data (RR series). An ECG signal is often disturbed, causing sudden changes in the measured heartbeat, either extremely fast or extremely slow. When this kind of artifact occurs, the RR series produces incorrect values. Thus, the RR series must be filtered using an efficient algorithm for filtering the RR interval series [21]. RR series were filtered using auto-adaptive thresholds in a moving window, with a window size of 5 samples, to detect artifacts. Lower and upper thresholds (m−2σ and m+2σ) were set for the first through fifth samples in the RR series, with m and σ denoting the mean and standard deviation of the entire series, respectively. Any samples from the first five samples below the lower threshold or above the upper threshold were substituted with the mean of the whole RR series. Then, the following thresholds were applied from the sixth sample until one sample before the last.
1.${\mathrm{RR}}_{i}<m_{5}-2\sigma_{5}$ and ${\mathrm{RR}}_{i+1}>m_{5}+2\sigma_{5}$2.${\mathrm{RR}}_{i}<0.75{\mathrm{RR}}_{i-1}$ or ${\mathrm{RR}}_{i+1}<0.75{\mathrm{RR}}_{i-1}$3.${\mathrm{RR}}_{i}>1.25{\mathrm{RR}}_{i-1}$
where
${\mathrm{RR}}_{i}$: the i-th sample of the RR series$m_{5}$: the mean of the previous five samples$\sigma_{5}$: the standard deviation of the previous five samples

Samples that satisfied one of the above conditions were marked as disturbances and replaced by the mean of the previous five samples of the RR series. To detect the disturbance in the last sample, we applied the following conditions:

1.

${\mathrm{RR}}_{i}<0.75{\mathrm{RR}}_{i-1}$

2.

${\mathrm{RR}}_{i}>1.25{\mathrm{RR}}_{i-1}$



The last sample was replaced by the mean of the previous five samples if it met one of the conditions above. Figure 2 and Figure 3 compare the RR series before and after filtering.

After filtering, the RR series data were resampled to 8 Hz using linear interpolation, resulting in an equidistant RR series. We then isolated the RR series into 64-s moving windows, with 512 samples per window and a 1-s moving period. Using Equation (1), the mean of the RR series in each window was calculated.
(1)$$M=\frac{1}{N}\overset{N}{\underset{i=1}{\sum }}{\mathrm{RR}}_{i}$$
where M is the mean of the RR series in the window and N is the number of samples in the window. To form the mean-centered RR series, Equation (2) was applied to each sample in the window, to subtract its mean.
(2)$${\mathrm{RR}}_{i}={\mathrm{RR}}_{i}-M$$

After that, the norm value S in every window was calculated by applying Equation (3) with the mean-centered RR series [22,23].
(3)$$S=\sqrt{\overset{N}{\underset{i=1}{\sum }}{\mathrm{RR}}_{i}{}^{2}}$$

The mean-centered RR series was then divided by the norm value by applying Equation (4), to form the normalized RR series.
(4)$${\mathrm{RR}}_{i}=\frac{{\mathrm{RR}}_{i}}{S}$$

In order to comply with the ANI formula, the RR series was downsampled in every window to 1 Hz.

Parasympathetic tone was correlated with ANI measurement, and this tone occurred only at high frequencies of the RR series. A Daubechies wavelet transform with four coefficients was used to band-pass filter the normalized RR series from 0.15 Hz to 0.5 Hz, yielding a time-scale representation of the RR series at different wavelet energy levels. The RR series with high frequency (RRHF) was then taken from the wavelet components 3 to 5. RRHF amplitude was expressed in normalized units (n.u.) and ranged from −0.1 to 0.1 n.u. [4]. Calculation of the area under the RRHF curve was done to assess the parasympathetic tone. To perform the calculation, the local maxima and minima in the RRHF curve were first detected. A plot of the upper line connected all maxima points, while a plot of the lower line connected all minima points, as shown in Figure 4. The 64-s RRHF window was then divided into four 16-s sub-windows. In each sub-window, the area under the curve (AUC) was calculated. ANI was calculated using the minimum AUC (${\mathrm{AUC}}_{min}$), which was the smallest AUC of the four, by applying Equation (5) [4,23].
(5)$$\mathrm{ANI}=\frac{100\;\times \;\left(5.1\;\times \;{\mathrm{AUC}}_{min}+1.2\right)}{12.8}$$

A denominator of 12.8 was used in the formula because the maximum possible value for total AUC was 0.2 n.u.×64 s =12.8 s. In addition, the coefficient and constant of 5.1 and 1.2, respectively, were used in the formula because both of them had been empirically determined in a general anesthesia dataset of more than 100 patients, in order to keep the coherence between the visual effect of respiratory influence on the RR series and the quantitative measurement of ANI [23].

ANI ranges from 0 to 100, similarly to EAPS, but with the opposite trend, as shown in Figure 5. A high ANI indicates high parasympathetic activity or low sympathetic activity, which is associated with low pain. Conversely, a low ANI indicates low parasympathetic activity or high sympathetic activity, both of which are associated with high pain. Next, by subtracting ANI from 100, the opposite value (100 − ANI) was obtained. Figure 6 shows that the opposite of ANI had the same trend as EAPS, so it was used as the input feature for training the deep learning models.

### 2.4. Deep Learning Models

The flowchart in Figure 7 summarizes the steps for building, comparing, and selecting the best deep learning model.

#### 2.4.1. Holdout

The holdout method was used to build the initial deep learning models and compare their performance, by manually selecting 60 patients for training, 10 for validation, and 10 for testing. Figure 8 illustrates the holdout method.

#### 2.4.2. Data Standardization

Scaling numerical input features to a standard range with centered data distribution can improve the performance of an AI algorithm. The input feature of the training, validation, and testing sets, which was the opposite of ANI (100 − ANI), were standardized prior to training the deep learning models. The standardization was begun by calculating the mean and standard deviation of the training set. For the training set, the input feature was subtracted with the calculated mean and divided by the calculated standard deviation, resulting in a mean of zero and a standard deviation of 1. Additionally, the validation and testing sets were also standardized in the same way, but the mean and standard deviation were calculated from the training set, not the validation and testing sets. Equation (6) was used to standardize the input feature.
(6)$$z=\frac{x-\mu }{\sigma }$$
where

z: The standardized datax: The data to be standardizedμ: The mean of the training setσ: The standard deviation of the training set

#### 2.4.3. Data Windowing

Each window of the input feature consisted of 10 data points, and each window moved for 10 s. As illustrated in Figure 9, the windows did not overlap because the moving period was 10 s. In every window, the EAPS at the current time (at t) was predicted based on the last 10 s of 100 − ANI (from t−9 to t), as shown in Figure 10. Table 1 shows the proportion of the data in the holdout method after windowing.

#### 2.4.4. MLP Model

Both the training and validation sets were assigned with a batch size of 32. After that, all batches of the entire data were brought into the MLP model. As illustrated in Figure 11, there were ten input nodes (based on ten time steps of 100 − ANI) and one output node in the MLP model. There were two hidden layers between the input and output layers, with 32 neurons each. The prediction curve did not fluctuate at all when less than 32 neurons were used in each hidden layer, so only a flat line could be generated as the prediction results. The prediction curve deviated farther from the target label curve when more than 32 neurons were used in each hidden layer or more than two hidden layers were used in the architecture, so the mean absolute error (MAE) increased. To maintain the scale of the data, batch normalization was applied in each hidden layer. In those two hidden layers and the output layer, sigmoid activation function was used. Before training the model, all labels were divided by 100, in order to comply with the sigmoid activation function (which ranges from 0 to 1). To make sure the model′s predictions were in line with the range of EAPS (0–100), the results were multiplied back by 100. The parameters were updated during training using Adam optimizer with a default learning rate of 0.001. Adam was chosen as the best optimizer for this MLP model because the other types of optimizers could not achieve the minimum validation loss. Following that, 30 epochs were run for training and validation. To see how well the model performed in predicting for different patients, the trained MLP model was used to predict the testing set.

#### 2.4.5. LSTM Model

An LSTM cell has the mechanism that is illustrated in Figure 12. First, the forget gate has the capability of selecting which information to discard and which information is kept in the memory. In this gate, a sigmoid function is applied to the combination between the previous hidden state, which is the short-term memory of LSTM, and the current input, to yield an output value between 0 and 1. The closer the output value is to 1, the higher the probability of keeping the information. Then, the previous cell state, which is the long-term memory of LSTM, is multiplied by the output from the forget gate. Next is the input gate, where there are sigmoid and hyperbolic tangent functions applied to the previous hidden state and the current input. The output from both functions are multiplied with each other, then the multiplication result is summed with the cell state outputted by the forget gate, to yield the updated cell state. After that is the output gate, where a sigmoid function is applied to the combination between the previous hidden state and the current input. The updated cell state is then passed to a hyperbolic tangent function. Last, the output from the hyperbolic tangent function is multiplied by the output from sigmoid function, yielding the new hidden state.

In the LSTM model used in this research, the training and validation sets were assigned with a batch size of 32. Next, the 10-s windows of 100 − ANI were brought into the LSTM model, which had two stacked LSTM layers between the input and output layers, as illustrated in Figure 13. Each LSTM layer consisted of 32 LSTM cells. The prediction curve did not fluctuate at all when less than 32 cells were used in each LSTM layer, so only a flat line could be generated as the prediction results. When an LSTM layer had more than 32 cells or when more than two LSTM layers were used in the architecture, the prediction curve fluctuated excessively and its position deviated farther from the target label curve, resulting in a higher MAE. To maintain the scale of the data, batch normalization was applied to each LSTM layer. The output layer was activated using sigmoid function. Additionally, Adamax optimizer was used to update the parameters during training, with a default learning rate of 0.001. The validation losses obtained from other types of optimizer did not differ significantly from the validation losses obtained from Adamax optimizer during the trials. The prediction results, however, only appeared as flat lines when other optimizers were used. In other words, for this LSTM model, Adamax was the only optimizer that could generate fluctuations in the prediction curve. Following this, 15 epochs were run for training and validation. More than 15 epochs of training and validation did not lead to any improvement, because the model never converged, which meant the validation loss always stayed close to the training loss. In other words, the validation loss curve never intersected the training loss curve.

#### 2.4.6. Model Selection

The prediction MAE and shapes generated from the MLP and LSTM models using the testing set were compared. The best model was determined based on the prediction MAE and the shape of the prediction curves.

#### 2.4.7. Seven-Fold Cross Validation

The best model was obtained. By using that model, 7-fold cross validation was applied, as shown in Figure 14. In this method, the testing set is the same as that used in the holdout method, so the remaining 70 patients were rotated between the training and validation sets. As for the validation set, 10 patients were randomly selected for each fold. Patients who had already entered the validation set in a fold did not enter the validation set in subsequent folds. As a result, they were included in the training set in other folds. To determine the best fold, the fold with the lowest prediction MAE was chosen.

## 3. Results

### 3.1. Analysis of EAPS Data

In order to check whether the distribution of EAPS data from the five medical doctors was similar, analysis of variance (ANOVA) was performed. The interval plot in Figure 15 shows that Doctor D gave a significantly lower EAPS, with a P-value of less than 0.05. Therefore, Doctor D′s EAPS was excluded. Accordingly, for every patient, there were four EAPS values from four medical doctors (Doctors A, B, C, and E), as shown in Figure 16. To train the deep learning models, the average of the four EAPS values was assigned as the target label.

### 3.2. MLP and LSTM Models in the Holdout Method

According to the holdout method, training and validation loss curves were obtained from both MLP and LSTM models after 30 and 15 epochs of training and validation, respectively, as shown in Figure 17 and Figure 18. Each loss value was multiplied backwards by 100, to match the real range of EAPS. Table 2 shows the training and validation losses of both models in the last epoch.

After training the models, the testing set (the other 10 patients) was predicted using the trained models. In Table 3, the MAEs between the predicted and the real EAPS are presented, and in Appendix A, the prediction results are visualized.

MLP had a lower overall prediction MAE than LSTM, as shown in Table 3. In addition, the prediction curves of MLP appeared to be more appropriate than those of LSTM, as shown in Appendix A. Therefore, MLP was selected as the best model to predict EAPS.

### 3.3. Seven-Fold Cross Validation with MLP

Table 4 shows the training and validation losses for all the folds in the last epoch. As with the previous method, the trained model of each fold was used to predict the testing set. The overall prediction MAE of each fold is also presented in Table 4.

The first fold had the lowest prediction MAE, and, therefore, was chosen as the best fold, according to the results shown in Table 4. Appendix A shows the prediction result of each patient from the first fold MLP model. As a final step, Table 5 calculates the overall MAE of the seven-fold cross validation method.

## 4. Discussion

As shown in Figure 17, the MLP model trained for 30 epochs converged very well using the holdout method. The validation loss curve approached and intersected the training loss curve in several of the last epochs when it reached the minimum validation loss. As shown in Figure 18, the LSTM model only required 15 epochs to reach low validation loss, unlike the MLP model. With the LSTM model, the validation loss curve approached, but did not intersect the training loss curve. As a result of this study, although LSTM is capable of memorizing sequences and selecting which information to memorize and to forget, it did not outperform MLP in predicting EAPS. With the MLP model, the prediction curves were better shaped. As shown in Appendix A, the MLP model’s prediction curves showed a better fluctuation shape, while the LSTM model’s prediction curves showed a stiffer fluctuation shape.

As illustrated in Appendix A, the predicted EAPS values were able to approximate the real EAPS values determined by the physicians most of the time, using either the MLP model in the holdout or the seven-fold cross validation method. Furthermore, the predicted EAPS also followed the real EAPS fluctuations. A decrease or an increase may occur in the prediction curves. However, it is not always possible to predict the fluctuations perfectly, similarly to the real EAPS. The prediction curves can increase or decrease earlier or later than the actual EAPS. In addition, the predicted EAPS was good at following slight fluctuations, but not very good at following sharp fluctuations. A sharp increase or decrease in the real EAPS was only followed by a slight increase or decrease in the predicted EAPS. Furthermore, in the beginning of the surgical procedure, the extremely high real EAPS was not followed by an extremely high predicted EAPS. Those imperfections in the prediction results occurred because the predicted EAPS was based on the ANI value, which refers to HRV, whereas the actual EAPS was not only based on HRV, but also the clinical experience of the physicians and the events that occurred during surgery, which means other factors remain that have not yet been covered in the input features of the current research.

Based on the actual and predicted EAPS values, the overall MAEs between the predicted and the real EAPS values were 2.460 ± 0.634 for the first fold MLP model and 2.490 ± 0.522 for the holdout method. According to the low MAEs, most of the predicted EAPS values were close to the actual ones. Furthermore, there were some points with a higher MAE between the predicted and real EAPS values, such as at the beginning and during sharp increases and decreases of the real EAPS. However, these parts of the data represented a minority, so the MAEs did not increase due to them.

Moreover, the overall MAE of the seven-fold cross validation method was 2.848 ± 0.308. The quality of the MLP model was quite consistent in all folds, due to its low standard deviation of 0.308, which means there were no big differences between the MAEs of all folds.

## 5. Conclusions

Providing continuous monitoring with objective pain assessment during surgery could help physicians to determine adequate analgesic dosages. In this study, we compared the performance of MLP and LSTM algorithms in predicting EAPS. MLP had a lower prediction MAE and a better shape of prediction curves compared to LSTM. In the future, further studies could consider other features such as photoplethysmography (PPG) and other kinds of continuous variables, so that the input features would be more complete in representing the factors that medical doctors use to determine EAPS, and the prediction results would be closer to the medical doctors’ gold standard.

## Figures and Tables

**Figure 1 sensors-22-05496-f001:**
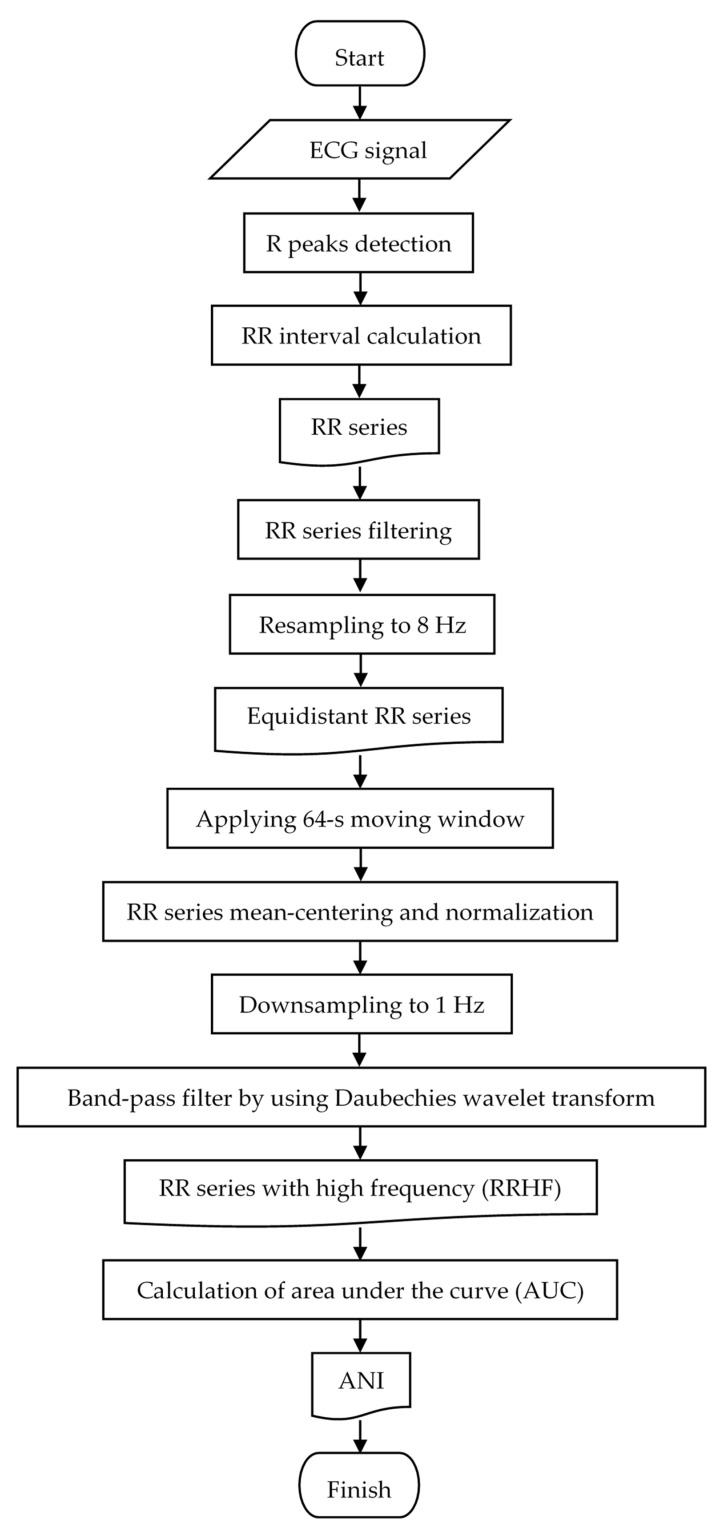
Flowchart of the ANI calculations.

**Figure 2 sensors-22-05496-f002:**
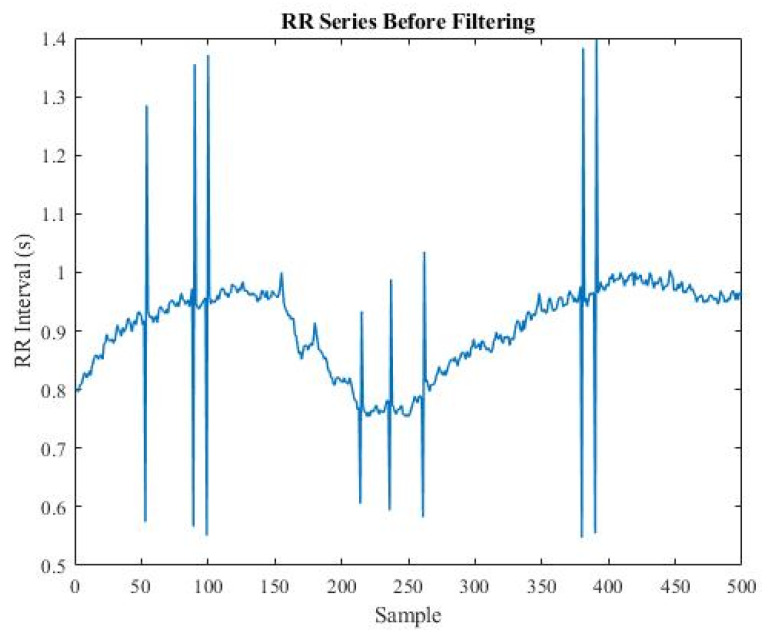
Plot of RR series before filtering.

**Figure 3 sensors-22-05496-f003:**
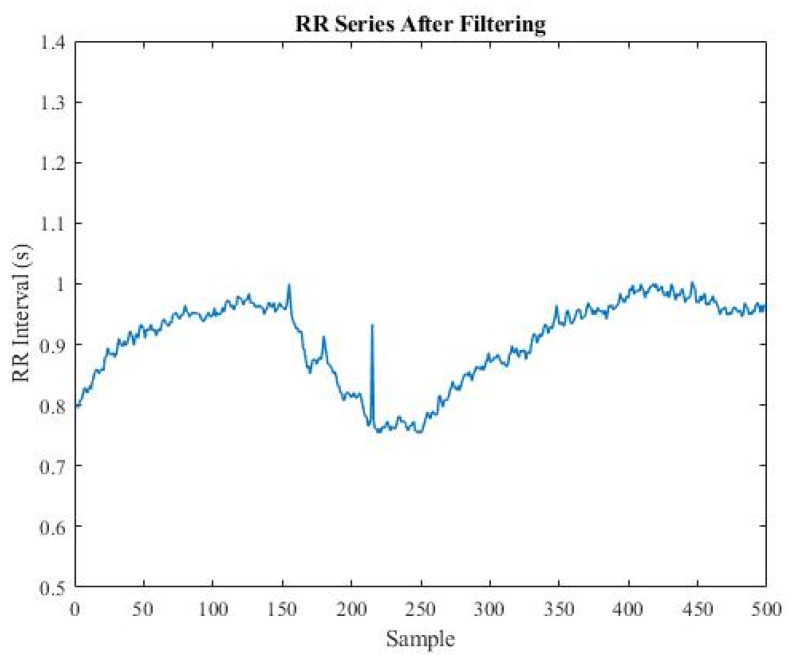
Plot of RR series after filtering.

**Figure 4 sensors-22-05496-f004:**
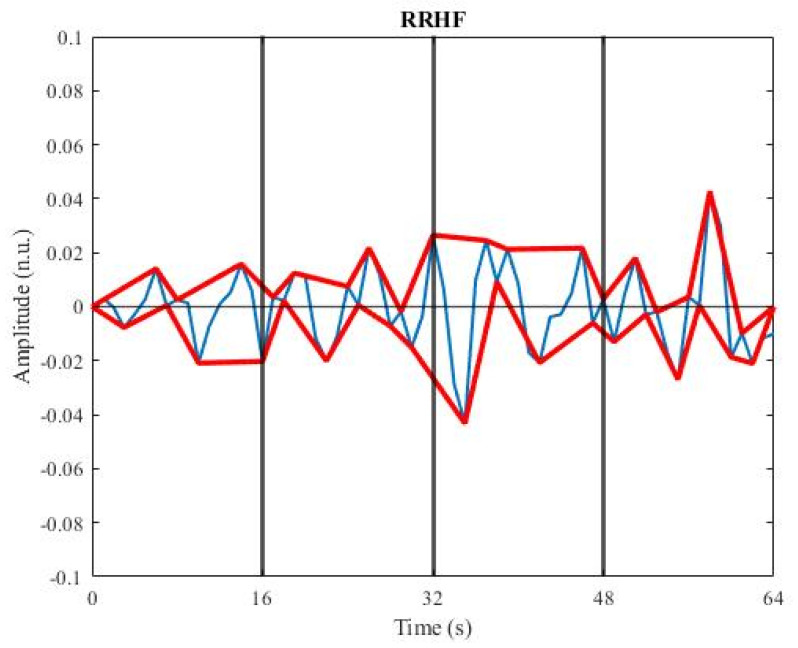
Plot of RRHF. Blue: RRHF curve; red: the lines which connect all the local maxima and local minima points in the RRHF curve.

**Figure 5 sensors-22-05496-f005:**
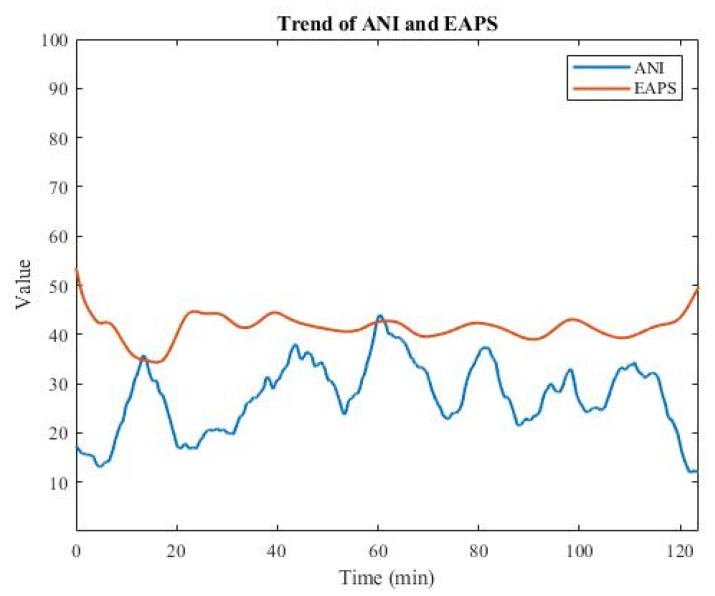
Plot of ANI along with EAPS.

**Figure 6 sensors-22-05496-f006:**
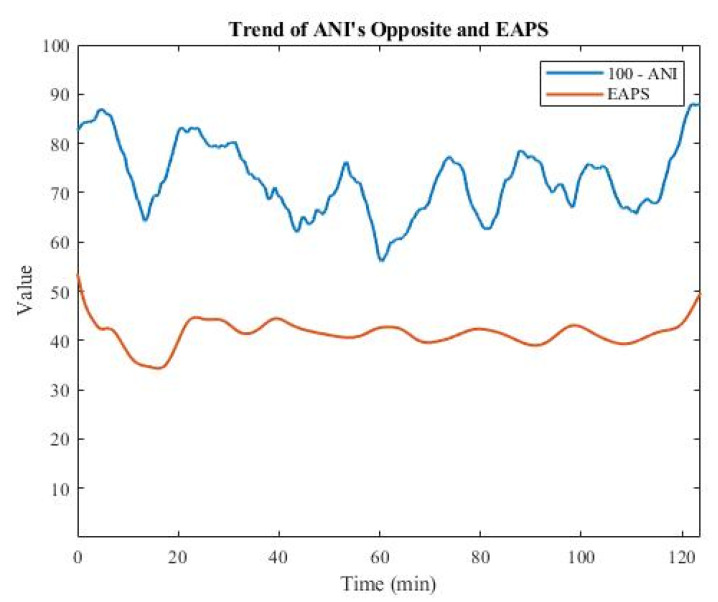
Plot of the opposite of ANI, along with EAPS.

**Figure 7 sensors-22-05496-f007:**
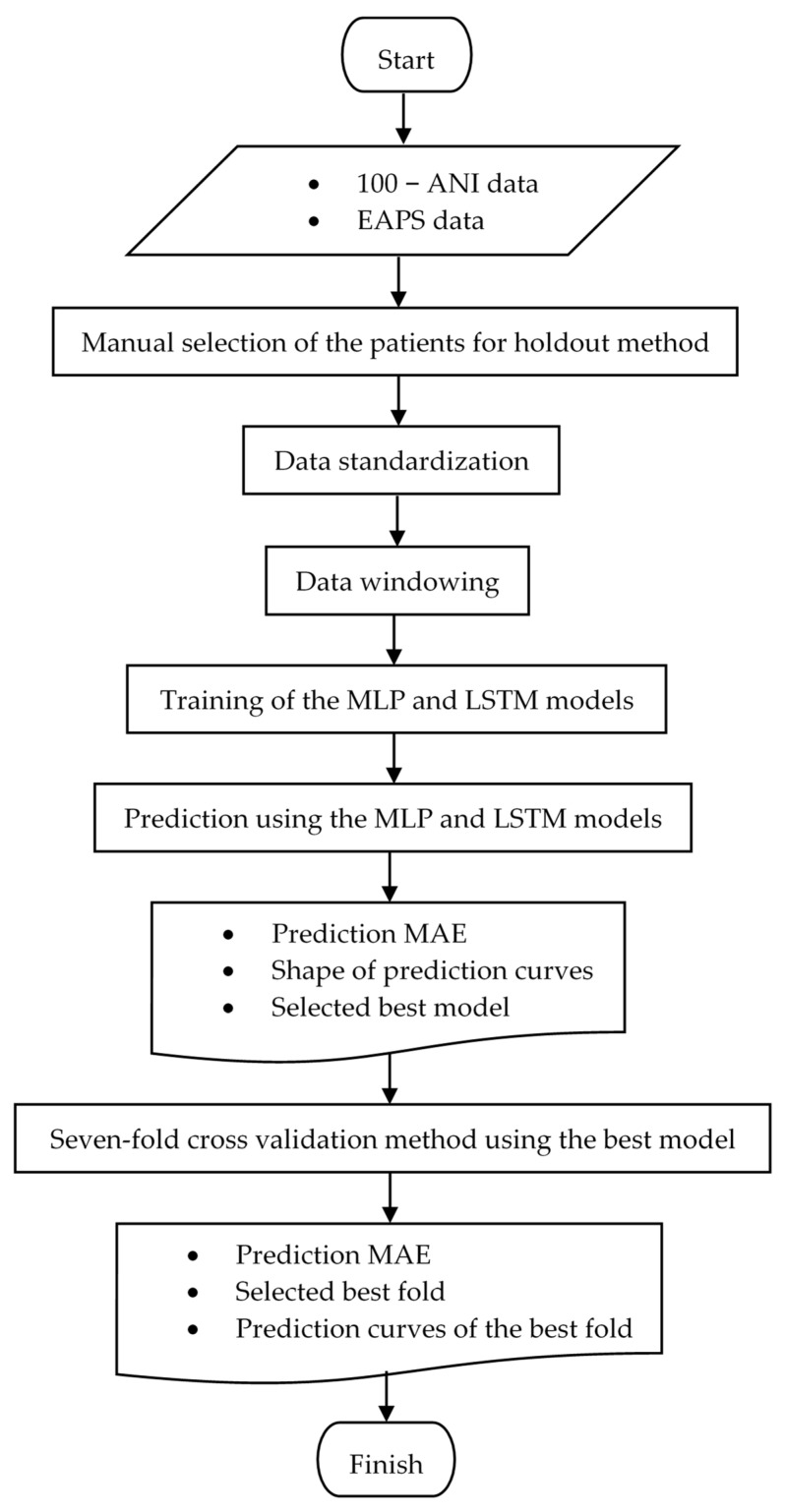
Flowchart of the deep learning model selection.

**Figure 8 sensors-22-05496-f008:**
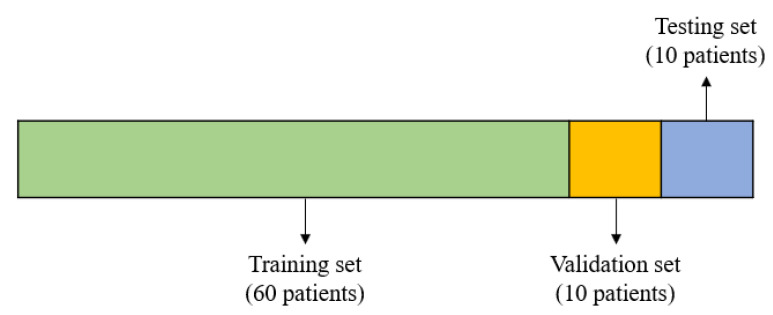
Illustration of the holdout method.

**Figure 9 sensors-22-05496-f009:**
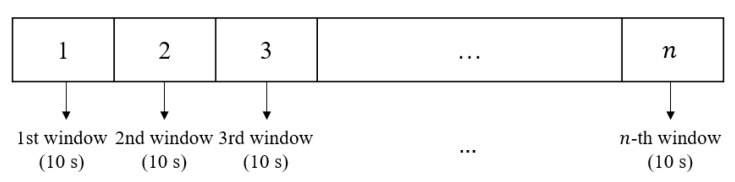
Illustration of 10-s non-overlapping moving windows.

**Figure 10 sensors-22-05496-f010:**
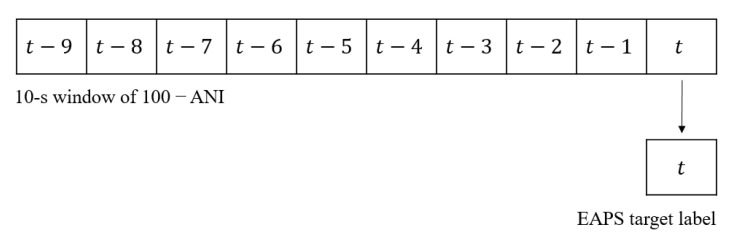
Illustration of the EAPS prediction.

**Figure 11 sensors-22-05496-f011:**
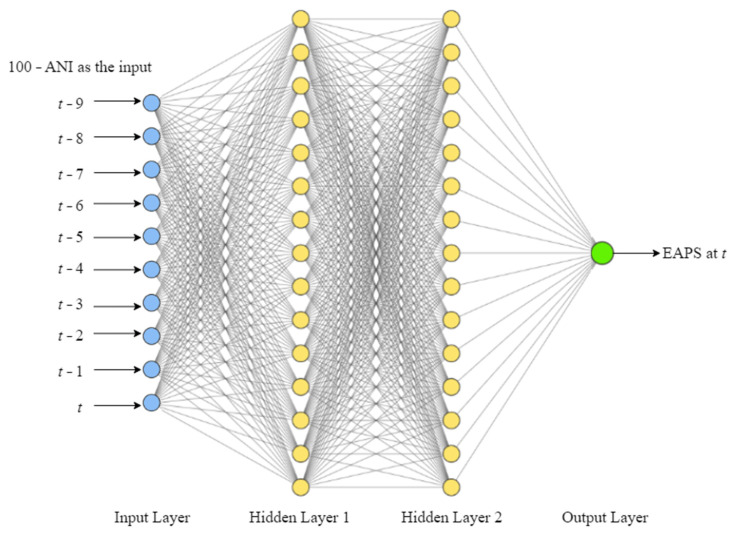
Illustration of the MLP model architecture.

**Figure 12 sensors-22-05496-f012:**
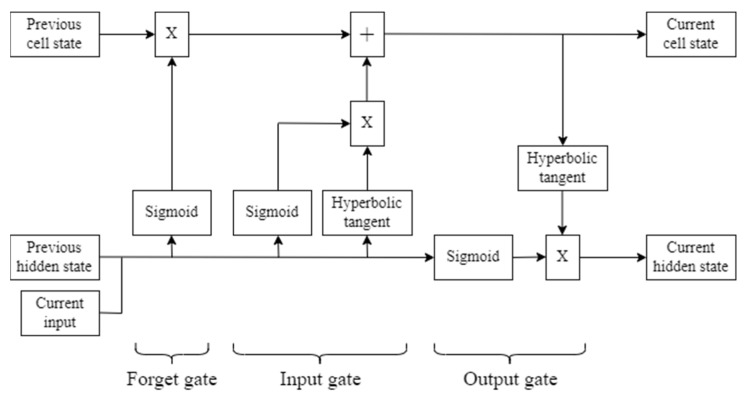
Mechanism of an LSTM cell.

**Figure 13 sensors-22-05496-f013:**
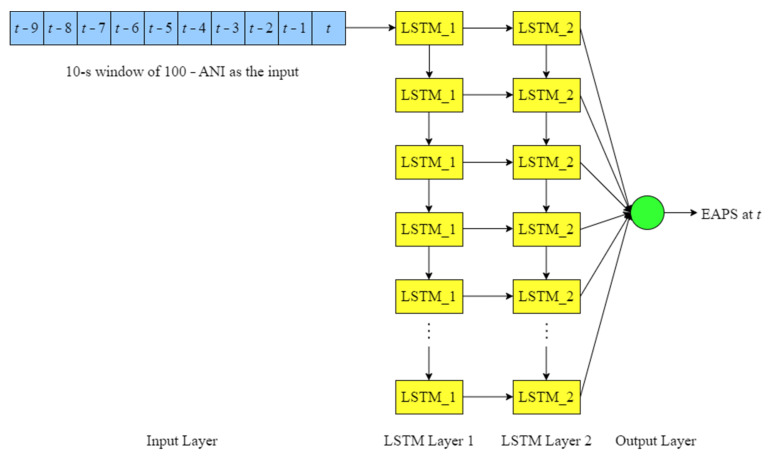
Illustration of the LSTM model architecture.

**Figure 14 sensors-22-05496-f014:**
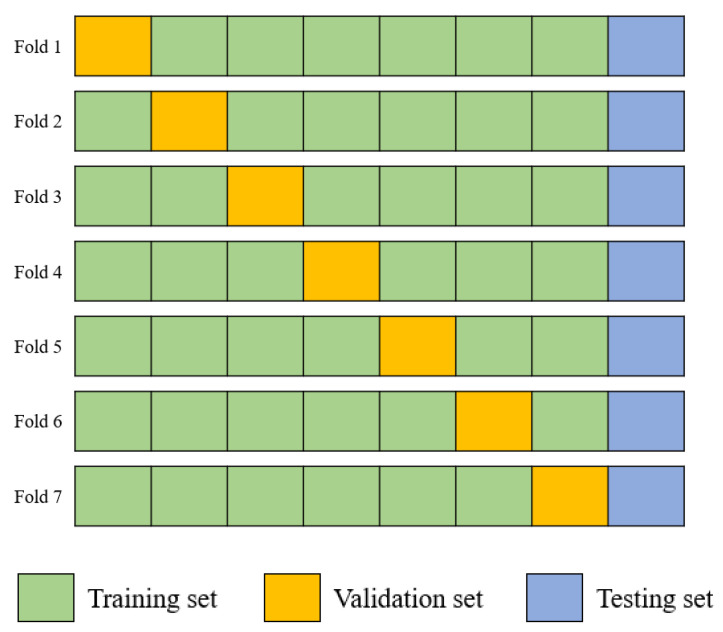
Illustration of the seven-fold cross validation method.

**Figure 15 sensors-22-05496-f015:**
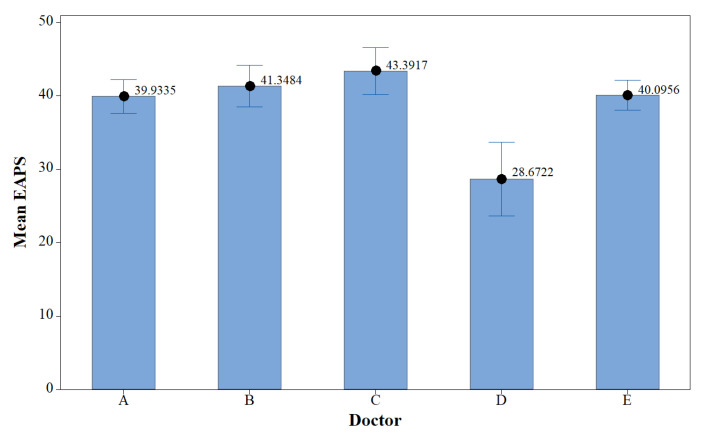
Interval plot of EAPS data distribution.

**Figure 16 sensors-22-05496-f016:**
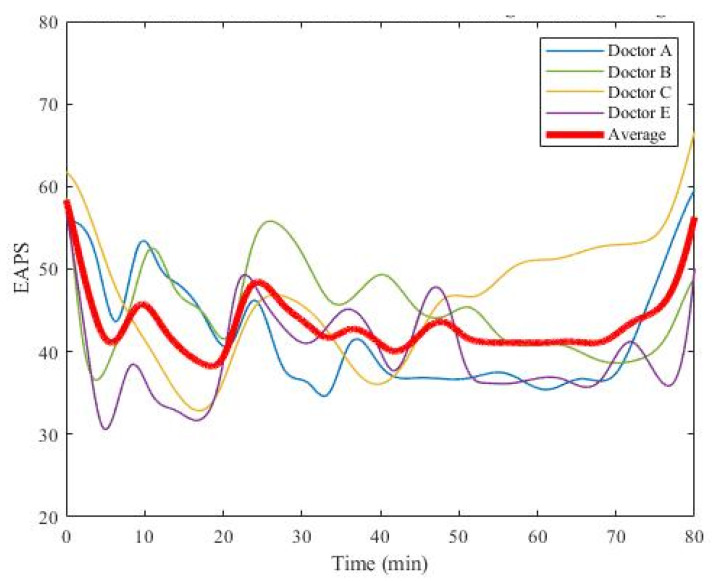
An average of the EAPS values from the four physicians.

**Figure 17 sensors-22-05496-f017:**
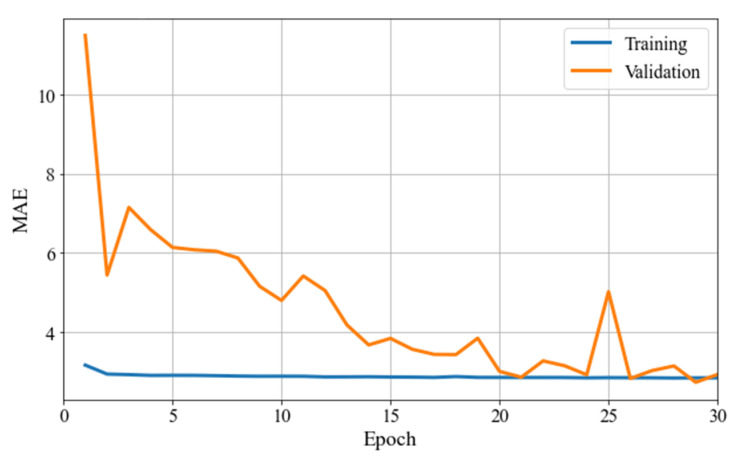
Training and validation losses of the MLP model in the holdout method.

**Figure 18 sensors-22-05496-f018:**
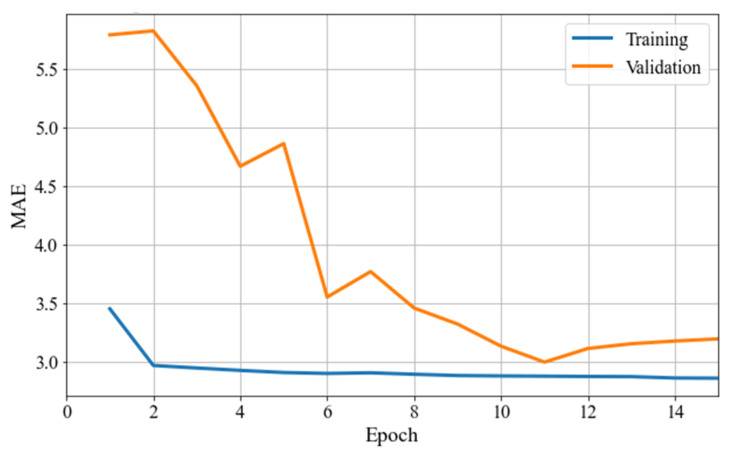
Training and validation losses of the LSTM model in the holdout method.

**Table 1 sensors-22-05496-t001:** Data proportion in the holdout method.

No.	Set	Number of Patients	Number of Windows
1	Training	60	47,554
2	Validation	10	9031
3	Testing	10	9374

**Table 2 sensors-22-05496-t002:** Training and validation losses of the last epoch in the holdout method.

Variable	MLP	LSTM
Training Loss	2.847	2.859
Validation Loss	2.933	3.194

**Table 3 sensors-22-05496-t003:** Prediction MAEs from both models using the holdout method.

Testing Patient	MAE of MLP	MAE of LSTM
1	2.716	3.238
2	2.622	3.493
3	3.422	2.271
4	2.118	1.886
5	2.127	2.768
6	1.875	1.948
7	1.933	2.242
8	2.735	3.076
9	2.214	2.671
10	3.138	2.731
Overall (mean ± SD)	2.490 ± 0.522	2.633 ± 0.542

**Table 4 sensors-22-05496-t004:** Training and validation losses in the last epoch in the seven-fold cross validation method, along with the overall prediction MAE.

Fold	Training Loss	Validation Loss	Overall Prediction MAE (Mean ± SD)
1	2.832	2.863	2.460 ± 0.634
2	2.793	3.195	3.075 ± 0.879
3	2.795	3.738	3.041 ± 0.673
4	2.869	2.640	2.542 ± 0.711
5	2.754	3.114	3.031 ± 0.948
6	2.804	3.799	3.209 ± 0.820
7	2.788	2.830	2.581 ± 0.711

**Table 5 sensors-22-05496-t005:** The overall MAE of the seven-fold cross validation method.

Fold	MAE
1	2.460
2	3.075
3	3.041
4	2.542
5	3.031
6	3.209
7	2.581
Overall (mean ± SD)	2.848 ± 0.308

## Data Availability

Data presented in the paper are available on request from the corresponding author J.-S.S.

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
