# Peer review of "Comparison of Deep Learning Algorithms in Predicting Expert Assessments of Pain Scores during Surgical Operations Using Analgesia Nociception Index"

_sensors, 2022, doi:10.3390/s22155496_

Round 1

Reviewer 1 Report

This manuscript compared two deep learning algorithms, multilayer perceptron (MLP) and long short-term memory, in predicting anesthesia of patients during surgical operations based on patients’ heart rate variability during the surgery. Although the topic of applying deep learning into medical field is an interesting topic, inadequate information has been provided to permit a clear understanding the novelty of the research, accuracy of the achievement results, reliability of the research. The authors should address the following:

1. The methodology of using the deep learning models should be clear. How are the models trained to predict the anesthesia of patients based on patients’ heart rate variability? How do characteristics of input data (heart rate variability) affect to the predict results?

2. The writing needs to be improved

3. As expressed in the abstract, “Analgesia nociception index (ANI), range from 0 to 100, is derived from heart rate variability (HRV) via electrocardiogram (ECG) signals for pain evaluation in a non-invasive way.”, why does it still need the deep learning models to predict anesthesia of patients during surgical operations as it can be obviously observed from ANI.

4. The contents of Multilayer perceptrons (MLPs) and Long-Short-Term Memory (LSTM) should be summarised in the introduction sections.

5. The differences between the architectures in Figures 15 and 16 should be shown in detail. Author should present the reasons for building these architectures?

Author Response

Thank you.

Reviewer 2 Report

This paper presents a comparison of two different but well known approaches of deep learning in the problem of expert assessments of pain scores prediction during surgical operation. For this purpose, the authors propose the use of the Analgesia Nociception Index (ANI) as input to two different prediction networks, a multilayer perceptron and a LSTM network to predict the EAPS score based on the patient's HRV during surgery.

Reviewer's comments
===================
The authors work on a very important problem that most aneasthesiologists face during patients' surgery. It is very important for them to estimate the level of pain in order to decide treatment plans and to see the outcome of these plans in order to make all the necessary ajustments in real time. However the assessment of pain can not be determined with the same criteria as it can be subjective or objective. So the main problem occurs under general anesthesia when the patients are unconcious and pain has to be determined and estimated objectively.
1. The description of Multilayer Perceptron as well as the LSTM (in a smaller degree) can be avoided and skipped by the authors, as the training procedure of the MLP is well known. Some references instead should suffice.
Line 141: the authors have described LSTM as a new "algorithm", when it is clear that it is a network architecture.
Lines 199-200: the authors claim that ANI is more sensitive for valuating pain in propofol-anesthetized patients. They should try to explain their argument in more detail and give more references that justify this argument.
Lines 249-251: My main concern is what is described in these lines. The authors have used five doctors that interpreted the notes or recordings provided by nurses to determine the EAPS. The doctors have plotted EAPS curves on recorded paper ** by hand **, and these drawings were later scanned and digitized into numerical data. Is this a common practice in order to collect data? To be honest, I haven't seen this kind of data collection (mixture of analogue and digital methodology), and I'm very skeptical on the quality of the data that all the experimental procedure relies on. The authors should give more detail on this and justify the quality (errors in measures etc) of the collected data.
Line 259: What do the authors mean by interval plot of EAPS Data Distribution?
Line 261: Do the included EAPS plots of figure 5 belong to one patient, or are these the average plots taken all the 80 patients into consideration?
Lines 313-314: The authors should include some references on this claim.
Line 327: The authors should explain in more detail equation 16 and in particular the parameters included therein.
Line 347: In figure 11 some text is cut, while Data is included two times (maybe a mistake?)
Line 378: The authors have used the current ANI value, along with the previous 9 time points in order to predict the current EAPS value. Have they tried to use only past ANI values (ommit the current ANI value)?
Section 2.4.4 Do the authors train their networks using all patients' data? This is not very clear from the paper.
Line 401: Should the authors write ...in line with EAPS (0-100) instead of 10?
Section 4: In this section, the authors should further discuss in more detail the findings included in lines 603-608. Why is this the case?

Overall the paper is well written, but needs some additions and a complete rewriting of the discussion section. Again, the authors in my opinion should explain scientifically, how the process of data collection that they have used, ensures solid scientific results.

Author Response

Thank you.

Reviewer 3 Report

Introduction

I suggest giving the background of previous work that has been developed. I feel that a lot of space is used in giving concepts that can easily be found in books.

Identify the shortcomings of previous techniques and form a critique that makes it known that you have a problem and need to investigate it.

Justify why you use Deep learning and not other techniques. What advantages does Deep learning offer you over other techniques?

Materials and methods

Perhaps a diagram illustrating the different stages and most important activities would be more understandable.

Some of the figures in this section should appear in the results section, as they are the data analysis.

In the methodology, focus on describing how you achieved your objective. Do not do data analysis.

Results

Put the results according to the activities described in the methodology. I think this would help to give a smoother understanding of the results.

I am not sure, but it would be better to report all the graphs in Figure 20 in the supplementary material. 

There are so many objects (tables and figures) I wish they could be reduced in number.

Author Response

Thank you.

Round 2

Reviewer 1 Report

The author addressed most of the reviewer's concerns except the second comment "The writing needs to be improved ". The writing needs to be improved before publication.

Author Response

Thanks for your comments.

The whole revised manuscript has carefully been checked and improved greatly by all authors, including one of authors, Dr. Maysam Abbod, who is currently the Reader in Brunel University London, UK and lives in UK over 30 years. He serves Editorial Board Members of several prestigious journals of MDPI publisher, such as Sensors and Electronics. He also serves of Board of Editors of Engineering Applications of Artifical Intelligence of Elsevier publisher. Dr. Abbod is not only checking the structure of this revised manuscript but also carefully checking English Language and Style in this time.

The attachment file is our revised manuscript. Thank you.

Reviewer 3 Report

Thanks for your response. The paper is still needing scientific structure.

Author Response

Thanks for your suggestions.

(1) We have rechecked our manuscript structure. Hence, we have moved the LSTM structure of Figure 1 in the introduction section to the Materials and Methods Section in Figure 12. We also re-draw the MLP structure in Figure 11 and LSTM structure in Figure 13 which are applied in our study for the details structures. Please see our revised manuscript of the page 11, lines 306-310 in Figure 11, page 12, lines 333-347 in Figure 12, and pages 12 & 13, lines 352-361 in Figure 13.

(2) Finally, the whole revised manuscripts has carefully been checked and improved greatly by all authors, including one of authors, Dr. Maysam Abbod, who is currently the Reader in Brunel University London, UK and lives in UK over 30 years. He serves Editorial Board Members of several prestigious journals of MDPI publisher, such as Sensors and Electronics. He also serves of Board of Editors of Engineering Applications of Artificial Intelligence of Elsevier Publisher. Dr. Abbod is not only checking the structure of this revised manuscript but also carefully checking English Language and Style in this time.

The attachment file is our revised manuscript. Thank you.
